# The Impact of Social Media on Employee Mental Health and Behavior Based on the Context of Intelligence-Driven Digital Data

**DOI:** 10.3390/ijerph192416965

**Published:** 2022-12-17

**Authors:** Rong Zhou, Zhilin Luo, Shunbin Zhong, Xinhua Zhang, Yihui Liu

**Affiliations:** 1Department of Development Studies, Faculty of Business and Economics, University of Malaya, Kuala Lumpur 50603, Malaysia; 2General Education Faculty, Chongqing Industry Polytechnic College, Chongqing 401120, China; 3School of Business, Minnan Normal University, Zhangzhou 363000, China; 4The Personnel Department, Suzhou Early Childhood Education College, Suzhou 215004, China; 5School of Business, Nankai University, Tianjin 300071, China

**Keywords:** employee social media use, employee psychology, employee mental health, health behavior, double-edged sword effect

## Abstract

With the rapid development and widespread popularity of the Internet, employee social media use at work has become an increasingly common phenomenon in organizations. This paper analyzes 105 related papers from the Social Science Citation Index in Web of Science through Scoping Review to clarify the definition and characteristics of employee social media use and the types of social media and summarizes the current research methods. Then, the reasons for employees’ willingness and refusal to use social media and the positive and negative effects of employee social media use on employees’ work attitudes, behaviors, and performance are discussed. Then, the mediating variables, moderating variables, and theoretical frameworks used in the relevant studies are described, and a comprehensive model of employee social media use is constructed. Finally, this paper indicates future research directions based on the latest research results in 2020–2022, i.e., improving research methods, increasing antecedent studies, expanding consequence research, and expanding mediating variables, moderating variables, and theoretical perspectives.

## 1. Introduction

With the rapid development and widespread popularity of the Internet, the use of social media by employees at work has become an increasingly common phenomenon in organizations [1]. Because of the unique advantages of improving the convenience of information transfer and work organization efficiency, companies prefer to incorporate typical social media such as WhatsApp, Zoom and MS Teams, and other Internet office systems into their daily management practices. The use of social media by employees in the workplace has a significant impact on the way employees interact and share information with each other and with the organization, so there have been extensive academic studies in recent years [2]. Studies on the factors influencing employees’ social media use have been conducted in different aspects such as individual, work, technology, and organizational contexts [3]. In terms of the consequences of the impact of employee social media use, this behavior can have positive effects on organizations, such as increasing employee job involvement and organizational citizenship behavior [4,5]. Meanwhile, the findings of Kwahk and Park [6] also indicated a significant positive effect on job performance. However, not all the findings are positive, and some scholars have expressed concerns. For example, if employees use social media for recreational and social purposes for long periods of time, it will significantly crowd out time resources for work, which will seriously harm the productivity of the company [7]. In addition, employees may also use the job connections brought about by social media use to seek new positions, thus increasing employee turnover [8]. Cao and Yu [9] showed that excessive use of social media by employees can cause a certain amount of psychological stress to employees and thus reduce job performance. This shows that this behavior has a positive impact on the organization as well as a negative effect. In addition, although some results have been achieved in academic research on employee social media use, there are still some problems. Chu [10] systematically analyzed the articles on the relationship between the use of social media and employee performance from 2009 to 2018 with the method of meta-analysis, but there are no recent studies from 2020 to 2022. Sun et al. [11] mainly focused on the dark side of social media use and emphasized the conflicts brought by employees’ use of social media, but there are no reviews objectively evaluating the consequences of employees’ use of social media from both positive and negative aspects. van Zoonen et al. [12] classified relevant literature according to individual behavior theory, social behavior theory, and communication theory, but no paper built a comprehensive model to clearly show the research background and frontier development in this field. Therefore, it is necessary to clarify the development and problems of employee social media use research through a systematic review so as to promote the deepening and development of this research area. In summary, in order to further promote the development of the research, this paper uses a systematic literature review method to sort out the definition, types, research methods, after-effects mechanisms, mediating mechanisms, and moderating mechanisms of employee social media use based on the review of existing research results and comprehensively elaborates the research lineage and cutting-edge development of employee social media use from 2010 to 2022 and constructs an integrated model. On this basis, the paper further identifies the key issues in the current study and possible future research directions.

## 2. Materials and Methods

Scoping review is a method of literature review that can be used to summarize and synthesize the current state of research in a field. Nowadays, employee social media use is becoming more and more popular, and its definition, causes, and post-effective, mediating, and moderating mechanisms on employees’ psychology have triggered many scholars’ research. Therefore, it is necessary to review and summarize the relevant research to clearly present the research findings in this area to researchers and to indicate the direction for future research.

The article sources were first identified. The articles in this study were derived from the results of the Social Sciences Citation Index (SSCI) in the Web of Science (WoS) database, which is considered one of the most authoritative academic databases in the social sciences because of its extensive coverage of the fields of business, management, and psychology. The SSCI library of Web of Science is more rigorous in the peer-review process, and the research papers covered are of high quality and high standards. The research path in this paper is as follows (Figure 1).

We, two independent reviewers, retrieved relevant existing literature from the social science citation index in the WoS database using keywords “employee social media use”. The articles reviewed were from the period 1 January 2010 to 31 July 2022. We began from January 2010 because Aichner et al. [13] mentioned that the definition of social media changed dramatically in 2010. Previously, social media was a communication platform used by a group of people with common interests to share and deliver information, whereas afterwards, a large portion of empirical literature focused on the idea that social media can create and share user-generated content [1], which implies a change in the research focus of social media. Our goal was to review the literature on the definitions and reasons for employees’ use of social media and the post-effective, mediating, and moderating mechanisms on employee psychology and to publish as large a sample of papers as possible in high-quality journals during the coverage period. Therefore, only English-language journals from the Social Science Citation Index were available. These two reviewers’ search results are both included. The number of papers is 105.

It must be acknowledged that our search may not have exhausted all articles published on the topic during the study period due to database unavailability, inappropriate search terms, or human error (oversight) in the search process. However, we believe we have found most articles on this topic in academic journals during our search. Therefore, this dataset allowed us to outline what was to be studied, the methodology, the theories used, and possible information gaps.

## 3. Results

### 3.1. Definition and Characteristics of Employee Social Media Use

In order to clarify the definition of social media use by employees, it is first necessary to specify what is meant by social media. Aichner et al. [13] mentioned that the definition of social media changed dramatically in 2010. Previously, social media was a communication platform used by a group of people with common interests to share and deliver information, whereas afterwards, a large portion of empirical literature focused on the idea that social media can create and share user-generated content [1], which implies a change in the focus of social media research. The results echo the findings regarding the progression of social media research that the definition of social media has changed over time from a social platform in the past to a tool for information aggregation [14]. As for the definition of employees’ social media use, scholars have given different explanations depending on the focus of the study. Treem and Leonardi [15] considered it as information transfer and share, and Mantymaki and Riemer [7] argued that this behavior could help employees communicate with the organization, including with supervisors, colleagues, and subordinates and other people within the organization, while Leonardi et al. [16] focused on the idea that this behavior could enhance management efficiency.

The characteristics of social media have profound implications for the study of the mechanisms of the effects of employee use of social media. The first is visibility. The visibility of social media means that employees’ background information, network location, real-time movements, and what they think are visible to almost everyone [17], which can create privacy breaches for employees [18]. The second is persistence. Persistence refers to the fact that information on social media does not expire or disappear [15]. This means that social media can accumulate massive amounts of information over time, making it difficult for employees to quickly locate targeted messages [18]. Furthermore, for the characteristic of persistence, social media is likely to retain employees’ historical records on social media, which can lead to privacy breaches of employees [11]. Third is interactivity. Chen et al. [19] found that the interactivity will allow supervisors or co-workers to communicate with employees at any time, thus interrupting their work [20].

### 3.2. Types of Social Media Used by Employees

Initial research did not delineate social media; for example, Charoensukmongkol [3] directly showed that using social media when working plays a positive role in both employee job satisfaction and job performance. However, as research progressed, some scholars found that in order to more clearly articulate the influencing mechanisms of employees’ social media use, it was necessary to differentiate the social media used by employees. There are several methods of classification.

In terms of the purpose of use, there are work purposes and social purposes [4]. Work purposes refer to this behavior mainly for work-related task activities; social purposes mainly refer to employees’ use for entertainment, leisure, and other non-work-related activities. Huang and Liu [2] argued that social media use for work purposes can better improve employees’ work efficiency, well-being, and job satisfaction. Social media use by employees for social purposes when working is often considered to be a massive waste of time and can reduce the productivity of the company [11]. However, there are some studies that suggest that employees’ socially-oriented social media use does not necessarily reduce organizational productivity. Song et al. [21] showed that work-oriented and socially oriented social media use are a complementary resource that can create synergistic effects and improve team and employee performance.

Social media can be classified into public and private social media. Public social media refers to the type of social media provided or identified by the organization in the workplace. Private social media refers to private social media used by employees when working. van Zoonen et al. [5] suggested that public social media can increase the effectiveness of organizational communication and employee engagement at work and reduce fatigue. However, research has also shown that the use of public social media disrupts employees’ attention, interrupts workflow, and increases work–life conflict [20]. Regarding private social media in the workplace, some scholars have argued that it can result in wasted employee time and reduced productivity [22]. Sun et al. [11], on the other hand, argued that there is no strong evidence supporting that employees’ use of private social media hinders work productivity; rather, many employees are able to maintain the energy and vitality needed to work effectively by using private social media to chat with family and friends, which can help them release the stress they face at work.

### 3.3. Research Methods of Literature on Employees’ Social Media Use

The existing literature uses two main types of research methods, namely qualitative interviews and quantitative questionnaires. A small number of scholars have used qualitative interviews. Charoensukmongkol [3] used interviews to ask employees about the effectiveness of their social media use and attended workshops with managers and employees to investigate which social media they used and to what effect. Ainin et al. [23], on the other hand, adopted semi-structured in-depth interviews with employees. Schmidt et al. [24], on the other hand, looked at how this social media connection among colleagues affects perceptions of organizational support and organizational spontaneity by surveying users about the number and percentage of total colleagues on social media. More studies have used questionnaires. Ellison et al. [25] were the first to design a social media use scale designed to measure individuals’ attachment to Facebook, primarily measuring the intensity, and later studies developed a Facebook utility scale based on this scale, including three different usage utilities: information utility, organizational utility, and networking utility [26]. This scale was later adopted by scholars such as Moqbel et al. [27]; Charoensukmongkol [3]; and Huang and Liu [2] and replaced Facebook in the original scale. In addition to measuring the utility, several studies have measured the frequency of use and purpose of use for employees’ social media use. For example, Leftheriotis and Giannakos [22] designed a scale that included the purpose, effect of use, and reasons, respectively. van Zoonen et al. [5] measured social media use by measuring the work-related use frequency of Facebook, Twitter, and LinkedIn to measure the frequency. In addition, as mentioned earlier, several scholars have classified employee social media use according to the purpose of use and type of media use and designed corresponding measurement scales. For example, Ou and Davison [28] classified social media use into use at work and use in daily life and measured the frequency. Zhang et al. [4], on the other hand, classified it into work-related and social-related social media use and measured them separately. Cai et al. [29] designed and used the Corporate Public Social Media Use Scale in their research survey. In addition, the most recent development on social media use at work scale comes from Cao and Yu [9], namely the Social Media Overuse at Work Scale, which divides social media overuse at work into two dimensions of excessive social activities at work, excessive hedonic and excessive cognitive use, which are measured separately.

### 3.4. Reasons Why Employees Use Social Media

The reasons for employees’ use of social media are divided into internal and external factors. Internal factors are based on employees’ own characteristics. Akman and Mishra [30] argued that female employees are more likely to use social media than male employees. Although the conventional impression is that younger employees are more enthusiastic about social media use than older employees, Leftheriotis and Giannakos [22] argued that employees will use social media regardless of their age when they are allowed to do so at work, meaning that the use does not decrease with age. In addition, it has been argued that individuals who prefer online chatting tend to use social media for communication and exchange [8]. Akman and Mishra [30] argued that employees’ personality traits influence their use of social media. The results indicated that employees with higher extroverted traits use social media more frequently for communication. Employees with higher neuroticism traits spend more time on social media use and are more willing to share personal information about their life and work on social media. Employees with higher accountability traits place a high value on meeting obligations and completing tasks on time and therefore are limited in their use of social media [30]. In terms of motivation for use, Zhang et al. [4] argued that employees use social media based on work purpose and social purpose. In terms of work purpose, employees use social media to accomplish the purpose of expanding work perceptions [31] and the purpose of sharing task information [32]. In terms of social purpose, compared to the complex social activities made in the past to establish new social relationships and maintain existing ones, employees’ use of social media can directly and simply interact and build social relationships with colleagues [2], which enhances employees’ social capital [33] and ultimately will facilitate employees’ access to social support and resources at work [34]. In addition, employees use social media in order to relax and take a break, and using social media after long working hours can reduce employees’ work fatigue.

External factors mainly come from the organization. Badea [35] argued that organizational culture may potentially limit or encourage individual employees’ social media use behaviors in the workplace. Building on this, Fusi and Feeney [36] argued that innovative organizational cultures encourage the behavior. In addition to organizational culture, organizational support can also have an impact. For example, Charoensukmongkol [3] found that colleague support and job demands were positively related to behavior, while supervisor support had negative influence, and Zhang et al. [4] later confirmed this view. Organizational environment also influences employees’ social media use. Tajudeen et al. [37] stated that employees are likely to use social media when an environment exists in the company where social media can be used. Tajudeen et al. [37] also confirmed that in an organizational environment, institutional safeguards are associated with behavior in a positive correlation.

While the above literature is about employees’ willingness, some scholars have studied the reasons why employees refuse. One reason is values conflict. Holtzblatt et al. [38] showed that cultural conflicts can prevent this behavior. Oostervink et al. [17] argued that before using social media in the workplace, employees were able to distinguish between business and private matters when communicating with colleagues. However, after that, the two are highly confused, thus creating a conflict in employees’ values, thus causing them to refuse to use social media. The second reason is work–life conflict. Because social media is installed on employees’ communication devices, employees can work anywhere at any time, which triggers the work–life conflict [39]. The third reason is privacy invasion. Xiao and Mou [40] argued that because social media contains massive amounts of personal information, this behavior may endanger employees’ personal privacy. For privacy protection, employees may refuse social media. Fourth, the use of social media may bring about information overload, communication overload, and social overload [9,41]. Overuse of social media triggers information, communication, and social overload, which leads to perceived exhaustion among employees, which in turn reduces their job performance [41].

### 3.5. Post-Effective Mechanisms of Employee Social Media Use

Empirical research has focused on the factors and the effects on work attitudes, work behaviors, and work performance. There is some disagreement among scholars about the after-effects and mechanisms, and it is necessary to integrate related studies to effectively promote the understanding of social media use in the workplace.

#### 3.5.1. Employee Attitude

Several studies have concluded the positive influence on employees’ attitudes. Charoensukmongkol [3] argued that employees’ social media use has an important role in reducing work stress. In addition, social media use can be effective in alleviating work–family conflict among employees [27], enhancing employees’ happiness and sense of belonging [6], and enabling employees to make higher organizational commitment [42], which ultimately helps to reduce employees’ willingness to leave their organization [4]. Not only that, but numerous studies have also confirmed that social media use plays an important role in increasing employees’ job satisfaction and reducing employee burnout [2,3,4].

However, it has also been suggested that social media use can negatively affect employee attitudes, including perceived stress, perceived anxiety, and emotional exhaustion. Regarding research on perceived stress, Chen and Wei [43] analyzed employees’ different social media use behaviors and found that perceived stress could be triggered because of information and social overload. Yu et al. [41] constructed a stressor–strain-outcome model and found that excessive use is a stressor, which triggers employees’ perceived stress. Cao and Yu [9] further investigated how employees’ excessive use of social media caused job stress and how perceived stress had an impact on job performance. Regarding the study of perceived anxiety, Wu [44] highlighted that social media is an emerging technology, and the use of emerging technology at work can cause employees to worry and be anxious about their jobs. In addition, Ayyagari et al. [39] noted that because social media can be used anywhere at any time, it requires employees to respond to work demands at all times, leading to anxiety. van Zoonen et al. [45] found that this behavior triggers emotional exhaustion in employees. Liu et al. [46] argued that when social media is commonly used in the workplace, supervisors and colleagues not only use it during normal work hours but may also continue to assign tasks or discuss work through social media during breaks, which leads to emotional exhaustion among employees. van Zoonen et al. [45] demonstrated that because social media can be used anytime and anywhere, it means that work invades life and triggers work–life conflict, and this conflict depletes their emotional resources, leading to emotional exhaustion [47]. Moreover, Bright et al. [48] found that privacy breaches associated with social media use can also lead to exhaustion and thus emotional exhaustion among employees.

#### 3.5.2. Employee Behavior

In terms of the impact on employee behavior, several studies have considered the positive impact on employee behavior. Bizzi [8] found that the use of social media is beneficial in increasing the level of knowledge sharing among employees and facilitating the cognitive absorption of employees at work [3,24]. Not only that, but van Zoonen et al. [5] revealed that this behavior can also enhance employee engagement and engagement at work. Bakker and Demerouti [49] found that this behavior can develop social capital by reaching out to a wide range of colleagues and supervisors in the organization and gaining their resources and support.

However, there are also studies that suggest the negative impact on employee behavior. Andreassen et al. [50] found that employees using social media in the workplace are likely to use social media for entertainment, which distracts employees from their work and leads to a decrease in work engagement and dedication. Some scholars have found that employees using social media may reach out to employees in other organizations to find and obtain new job opportunities [8], thus increasing the likelihood of employees leaving their current organizations. Knowledge hiding is another negative behavioral outcome. Oostervink et al. [17] found that this behavior hinders knowledge sharing because employees cannot handle work–life conflicts. Sun et al. [51], on the other hand, argued that information overload and privacy invasion hinder employees’ knowledge sharing when using social media. Chen et al. [52] argued that social media information visibility puts employees in a position to hide knowledge and creativity for privacy protection. In addition, the use of social media blurs the work–life boundaries of employees and causes work–life conflict, so employees make countermeasures accordingly by developing personal boundary management strategies. Banghart et al. [53] found that some employees would turn off social media after work for work–life balance, while some employees would reduce their personal information disclosure on social media in order to distinguish between work and life. Another strategy is strategic responsiveness [54]. Because of the prevalence of social media, employees can respond immediately to communication needs from social media at any time and any place, which creates a strong sense of work overload for them [55]. To avoid feelings of work overload, some employees use strategic response, i.e., actively responding to communication demands from social media during work hours while prioritizing life matters during non-work hours, which increases their job autonomy [20,56]. In conclusion, social media is positive for employee behavior if it is used appropriately at work [31,57]. However, social media overuse may have negative effects [58].

#### 3.5.3. Employee Performance

Performance is considered to be one of the most important indicators of employee performance. Employees’ use of social media has a positive impact on job performance. In terms of personal performance, Moqbel et al. [27] concluded that social media use is beneficial in improving employees’ job performance, and this was confirmed by study, which concluded that this behavior makes it easier and more convenient to get advice from colleagues, thus contributing to improving employees’ productivity. In subsequent studies by scholars, findings have generally supported the positive effect on work performance [59]. In terms of team performance, Cao and Ali [60] argued that effective use of social media in teams can enhance team performance, especially innovative performance. Building on this, Ali et al. [61] classified the uses of social media into social, cognitive, and hedonic uses for their study, and the results showed that social media use facilitates collective innovation effectiveness by allowing team members to develop credible knowledge coordination systems within the team, ultimately improving team innovation performance. In addition to job performance and team performance, research has found that this behavior can also enhance employee agility performance [29]. In addition, social media use has also been shown to improve managers’ and employees’ decision-making performance and help improve the quality of decisions [8].

However, some other studies have argued that social media use reduces employee performance. Koch et al. [62] found that employees perceive the workplace as formal and strict, while social media is usually informal and flexible. When using social media in organizations, employees often have difficulty integrating it with a rigorous workplace and often use social media to entertain themselves in serious work situations, resulting in lower job performance [63]. Other scholars argue that excessive use leads to work intrusion into the life domain, which triggers conflicts [64], and these conflicts trigger employees’ fatigue, which leads to lower job performance. Cao and Yu [9] found that the excessive use triggered technology–work conflicts, which increased their psychological stress and decreased their job performance.

### 3.6. Mediating Mechanisms

The effects of social media use on employee performance, behavior, and attitudes are often not direct, and studies have examined the mediating mechanisms of social media use outcomes in two main areas: employee perspective and organizational perspective.

Employee perspective is at the level of employee attitudes. Studies have examined the mediating role of job satisfaction, online social capital, organizational commitment, work–family conflict, and work–technology conflict. For example, Charoensukmongkol [3] considered job satisfaction as the mediating variable, but his opinion was questioned by some scholars [27]. Huang and Liu [2], on the other hand, pointed out that employees’ social media use can generate a kind of online social capital, which refers to a kind of social capital generated and maintained mainly through the Internet. In addition to this, Zhang et al. [4] concluded that organizational commitment is a mediating variable and that different purposes of social media use positively affect employees’ job satisfaction and negatively affect their intention to leave their jobs by affecting their organizational commitment. However, some studies have also argued that employees’ social media use can negatively affect job outcomes by affecting some organizational variables, such as some scholars arguing that social media use can cause employees’ work–family conflicts or technical work conflicts, which can lead to job burnout and reduce job performance [59].

Organizational perspective is at the level of social interaction. Studies have examined the mediating role of social connectivity, task dependencies, knowledge acquisition, knowledge shareability, and communication effectiveness. Sun et al. [51] argued that the use of social media makes it easy to develop relationships among employees and leads to a “feeling of closeness to other members” and that this social connectivity may reduce employee turnover and improve employee performance. Task dependency refers to the degree of interaction and coordination between different tasks at work, and Pitafi et al. [65] suggested that this behavior facilitates the processing and refinement of information, increases the efficiency of communication between employees, and facilitates interdependence between tasks, and a high degree of task dependency leads to positive work outcomes. Interactive memory system refers to how team members share their knowledge and expertise, and it is an important factor that affects team performance. Cao and Ali [60] stated that social media facilitates the interactive memory system by improving communication, coordination, and trust among team members. Knowledge acquisition refers to work-related knowledge and acquired experience. Sun et al. [18] argued that social media use triggers knowledge acquisition so as to influence individual employees’ innovation performance and job performance. In summary, social connectivity, task dependencies, interactive memory systems, and knowledge acquisition act as mediating mechanisms between social media use and employee performance, team performance, and innovation performance. In addition to this, this behavior can also enhance employee job satisfaction, engagement, and well-being by increasing knowledge shareability and communication effectiveness [6].

### 3.7. Moderating Mechanisms

The strength of the impact of social media use on employee performance, behavior, and attitudes is often influenced by a number of other variables. Research on the moderating mechanisms of social media use has focused on both employee factors and work situation factors.

In terms of employee factors, employee job satisfaction, online social capital, IT competency, and pre-performance have been identified as important moderators of influencing social media use. For example, it has been shown that job satisfaction positively moderates the relationship between the intensity of social media use and employees’ cognitive absorption and job performance [3,27], and the higher the job satisfaction of employees, the stronger the effect of social media use on cognitive absorption and job performance. Online social capital refers to the social capital acquired by employees using social media [66]. Huang and Liu [2] argued that online social capital has a moderating role in the relationship between social media use and employee performance. The higher the employee’s online social capital, the stronger the ability to use social media to handle work tasks and the stronger the impact of social media on employee performance. Similar to online social capital, employees’ IT competency, i.e., the ability to use social media technology, is also believed to act as a moderator in the relationship between social media use and job performance. Pitafi et al. [67] concluded that the higher the IT competency of employees, the greater the impact of social media use on job performance. In addition, pre-performance is a manifestation of employee self-efficacy, and employees with high pre-performance are perceived to be more efficient and autonomous. Lu et al. [68] concluded that the higher the pre-performance of employees, the greater the role of social media use in improving job performance.

In terms of work context factors, workplace virtuality, task relevance, and task ambiguity were identified as important moderators of influencing the role of social media use. Wu et al. [69] argued that workplace virtualization leads to geographically dispersed employees, and employees working in such teams must rely on social media technologies to accomplish their work in a virtual work environment; thus, virtualization plays a moderating role between social media use and performance. Task relevance requires employees to frequently exchange or seek knowledge, and task ambiguity requires employees to deal with unstructured business problems or new issues. Both high task relevance and high task ambiguity positively influence employees’ social media use to achieve higher levels of performance. Therefore, task ambiguity and task relevance play a moderating role between employees’ social media use and performance. In addition, organizational social media policy refers to the guidelines and regulations made by an organization regarding employee social media use within the workplace [70]. Employees’ social media use is directly influenced by organizational social media policies, and some scholars argue that lenient or instructive organizational social media policies increase the likelihood that employees will use social media to obtain the resources they need to do their jobs, which in turn leads to improved job performance [70]. Thus, organizational social media policies play an important moderating role in the relationship between employee social media use and job performance.

### 3.8. Theoretical Perspectives

In explaining the influence mechanism, studies have involved different theoretical perspectives. This paper summarizes the relevant theoretical foundations that influence employees’ social media use and the relationship between social media use and employees’ job performance, job behavior, and attitude from three perspectives: mass communication perspective, individual motivation perspective, and interpersonal interaction perspective, respectively.

The mass communication perspective mainly includes the media richness theory and the use and satisfaction theory. Based on media richness theory, Kaplan and Haenlein [1] argue that social media are more effective than any previous media in resolving ambiguity and reducing uncertainty and therefore can improve the effectiveness of organizational communication and increase communication satisfaction. Not only that, but Huang and Liu [2], also based on media richness theory, argued that this behavior provides a variety of communication opportunities for the development and maintenance of employee relationships, and employees may use social media to find valuable information about their less-familiar colleagues. This behavior enhances job satisfaction by improving relationships with co-workers on the one hand and enhances organizational productivity and employee performance on the other. Luo et al. [42] showed that this behavior increases employee satisfaction. In addition to this, van Zoonen et al. [71], based on use and satisfaction theory, found through a survey study that this behavior is positively associated with engagement by increasing the accessibility of organizational information and effective communication.

The individual motivation perspective mainly includes belongingness theory, resource conservation theory, and social cognitive theory. Belonging theory suggests that belonging is one of the basic needs of individuals. Grieve et al. [72], based on belonging theory, argued that using social media such as Facebook may provide opportunities to develop and maintain social connections, which are associated with lower depression, lower anxiety, and higher subjective well-being, among other positive psychological outcomes. Resource conservation theory suggests that individuals always want to be happy and successful at work, and in pursuit of such a situation, employees acquire and protect valuable resources that help them construct and maintain the individual characteristics and social status needed to achieve success. Social cognitive theory suggests that a person’s behavior is influenced by individual cognitive levels and social factors and is often used to help understand individual motivation and behavior in different situations. Based on social cognitive theory, Kwahk and Park [6] argued that this behavior can enhance employees’ awareness of organizational decisions and the knowledge required for their work so that employees can update their knowledge base in a timely manner and participate more actively in the organization’s decision-making process, thereby increasing employee engagement and performance.

The interpersonal interaction perspective mainly includes social capital theory, social support theory, social penetration theory, and organizational commitment theory. Based on social capital theory, Cao and Yu [9] suggested that social media use can nurture employees’ social capital, which in turn promotes knowledge transfer, and that both social capital and knowledge transfer contribute to better job performance. Social support theory suggests that good work attitudes and work relationships can lead to more positive employee performance in the workplace. Based on social support theory, Moqbel et al. [27] showed that social media can promote job satisfaction by helping employees to achieve work–life balance and reduce work–family conflict. Social penetration theory describes the formation, maintenance, and ending of intimate relationships, and the theory suggests that self-representation is necessary for the formation and development of intimate relationships. Based on social penetration theory, Luo et al. [42] argued that due to employees’ more personalized, social, and informal activities on social media, there is a higher level of self-representation and expression, which positively affects employees’ organizational affective commitment. Organizational commitment is an important reflection of employees’ attitudes toward the organization and has a profound impact on employee behavior. Based on organizational commitment theory, Zhang et al. [4] concluded through an empirical study that this behavior can reduce employees’ willingness to leave their organization by increasing employee engagement and organizational commitment, which increases employee job satisfaction.

## 4. Discussion

The previous section constructs an integrated model of employee social media use, and this section describes the latest research findings from 2020–2022 based on this model, indicating future research directions.

### 4.1. Improve Research Methodology

There are certain problems with both research methods, whether qualitative interviews or quantitative questionnaires. One is sample bias. Because interviews are generally limited to a few companies, a few industries, and a few dozen employees, while the sample size of the questionnaire survey reaches up to several thousand, bias in the sample results is inevitable. The second is problem is that of being too traditional. These two methods have been used for decades. Nowadays, big data technology is widely used not only in natural sciences but also evolving in social science research. In Meng et al. [73] study on live e-commerce, they used Python technology to crawl data for the first time and then analyzed it with text analysis. Hence, whether or not big data technology can be used for employee social media usage, this update of research methodology deserves further study.

### 4.2. Increase Reason Studies

An analysis of the literature on the reasons for employees’ social media use for the three years of 2020–2022 revealed that internal and external factors have been expanded by relevant scholars. In terms of internal factors, Wei et al. [74], based on self-determination theory, found that three basic psychological needs of employees (i.e., need for competence, need for autonomy, and need for relatedness) influence employees’ social media use. Yee et al. [75] found that motivation, opportunity, and competence factors promote employees’ social media platform use from the employees’ perspective. In terms of external factors, Pekkala and van Zoonen [76] found that organizational commitment, social media training, and prior social media experience were the reasons. Schaarschmidt and Walsh [77] found that the importance employees place on corporate reputation determines how they use social media because employees are generally perceived as representatives of their employers on social media, and employee dynamics on social media shape the employer’s reputation. Several scholars have discussed the reasons from both internal and external perspectives. Oksa et al. [78] found that the intrinsic motivations for millennials include employees’ personal choices and their interest in discussing niche areas, and the extrinsic motivations are related to the organization’s work culture and personal brand. In addition, they found that millennials have experienced a major technology boom, are natives of the Internet, and are adept at using social media, but they have also experienced faster technological change and more burnout than previous generations. Zhang et al. [79] found positive effects of creative self-efficacy and perceived usefulness and that the use of leaders weakened employees’ use of social media. Lee [80] was the first to analyze the impact of personal (enjoyment, venting negative emotions, and self-improvement), interpersonal (bonding and bridging relationships), and organizational (organizational-employee relationships and perceived external prestige) factors by using the theoretical framework of the Social Ecological Model (SEM). In summary, the reasons for employees’ use of social media are mainly internal and external factors. Relevant studies in the last three years are still based on this category for further expansion. Future studies can start from a more novel perspective. For example, internal factors should be taken into account, such as employees’ religious beliefs and cultural background, while external factors should also be taken into account, such as colleagues’ competitive consciousness and Internet penetration rate.

### 4.3. Expand Post-Impact Mechanism Studies

In 2020–2022, further progress has been made in research on the impact of employee use of social media on employee attitudes. In terms of positive impact, Zhou and Mou [81] proposed a cross-level moderation model based on self-determination theory and found that this behavior positively influenced employees’ relational energy and spirituality. Zheng and Davison [82] concluded that work-related social media use increased employees’ affectionate identification with the team and enhanced the sense of teamwork. Furthermore, Liu et al. [83] concluded that this behavior increases employees’ enthusiasm for work. Specifically, the use of work-related social media increased employee enthusiasm by reducing obstructive stressors, while the use of social-related social media increased employee enthusiasm by reducing challenging stressors. Cheng and Cho [84], on the other hand, found through a questionnaire survey of some workers in the U.S. hospitality industry that this behavior was a break activity that restored their enthusiasm. In terms of negative effects, Tandon et al. [85] argued that this behavior creates a dark-side outcome, a psychological state of fear of missing out, which leads to job exhaustion. Taborosi et al. [86] conducted a questionnaire survey in western Balkan countries and found that prolonged use of social media reduces job satisfaction, while the use of multiple social media can seriously jeopardize organizational commitment. Tandon et al. [87] found that this behavior induced voyeurism and fear of missing out and compulsive use of social media among employees. Nam and Kabutey [88] noted that this behavior enhances the impact of emotional labor on burnout, which leads to emotional exhaustion. Whelan et al. [89] found that this behavior amplified the effects of information overload on social media fatigue. The research in the last three years has increased the variety of attitudes and achieved research in multicultural contexts, undoubtedly pointing the way for future research.

The 2020–2022 research on employee use of social media influencing employee behavior was further developed, mainly by increasing the types of behaviors and adding mediating variables. In terms of positive effects, the effect on employee agility was added. Zhang et al. [90] concluded that this behavior influenced employee agility through the moderation of innovation culture. Wei et al. [91] argued that this behavior increases employee agility through meta-knowledge. Pitafi et al. [92], on the other hand, argued that it enhances employee agility through job expertise. In addition, studies on the impact on employees’ innovativeness have been added. Wang et al. [93], based on social exchange theory, found that this behavior positively affects employees’ creativity through leadership member exchange. Cheng et al. [94] found that this behavior increased work engagement and thus positively influenced employees’ innovation performance. In terms of knowledge sharing, recent literature has further investigated the mechanism of action. Yang et al. [95] concluded that the visibility of social media increased employee knowledge sharing based on the visibility characteristics of social media. Organizational identity was added as a mediating variable. Yue [96] argued that the need for after-hours corresponding social media communication about work enhances organizational identity and thus increases employees’ work engagement, which is consistent with the findings of Oksa et al. [97] and Men et al. [98]. Other behavioral studies have also been added. For example, Nivedhitha and Manzoor [99] suggested that this behavior can increase social connections in the workplace and thus reduce cyber-slacking. Moqbel et al. [100] suggested that it increases job satisfaction and reduces job stress, thus decreasing turnover intentions. In addition, Cai et al. [101] concluded that it accelerates the integration process of new employees into the group by building interpersonal relationships. The negative effects have been further studied. In terms of knowledge sharing and concealment, the literature in the last three years has expanded the mechanism of action studies. Zhang et al. [79] found that the more employees use social media, the more detrimental knowledge sharing is because the more employees communicate on social media, the better they feel, and the less they share knowledge. Too much close collaboration and contact can instead create conflicts among employees and discourage knowledge sharing. Ma et al. [102] found that work-related public social media use inhibited employees’ knowledge hiding, and social-related ones promoted employees’ knowledge hiding. In terms of employee turnover, Tang et al. [103] proposed the novel hypothesis that employee social media use increases emotional exhaustion and exacerbates supply chain professionals’ intention to quit, but work–life balance suppresses the effect. Regarding whether it increased employees’ work engagement, Yue [96] found conflicting results that work-related social media use after working enhanced organizational identity, thereby increasing work engagement and leading to more work–family conflict, which in turn decreased work engagement. These three years have also added to other findings of negative effects. For example, Tandon et al. [85] suggested that social media-induced fear of missing out can lead to uncivil behavior in the workplace. Nusrat et al. [104] suggested that employees using social media will pretend to use social media for work but actually for pleasure, thus causing cyber-slacking. Huang and Fan [105] found that employees compare themselves with their social media friends from the novel perspective of social comparison. If others are found to be superior to them, the employee indulges in low self-esteem, and work performance decreases. The more enthusiastic the employee is about social media, the more likely he or she is to be in this situation. In addition to further in-depth studies based on the original framework, studies in the last three years have added behavioral categories and mediating variables so that future studies can propose new dependent or mediating variables.

From an analysis of the relevant literature for the last three years of 2020–2022, it is easy to find that the latest research reveals the corresponding mechanisms of action, namely through mediating and moderating variables. Ma et al. [106] found that the behavior can positively affect work productivity and emotional maintenance, thus influencing employee performance, and that frequency of use and personal characteristics may moderate this process. Chen et al. [107] found that using internal or external social media for work purposes could improve employee performance; using external social media for social purposes could also improve employee performance because of reduced fatigue, but using internal social media could lead to a decrease in performance because of distraction from work tasks. Zhao et al. [108] concluded that this behavior can positively affect employee performance by increasing work engagement and reducing work interruptions. Wu et al. [109] found that employees’ use of social media triggered job stress and that challenging stress contributed to employee performance, while hindering stress had a negative effect. Jia et al. [110] found that this behavior positively influenced knowledge acquisition, thus promoting task self-efficacy and creativity, which in turn improved the job performance of construction managers. Chen and Wei [111] found that this behavior facilitated leader-member and team member communication, thereby positively influencing employee performance. Saleem et al. [112] found that excessive use affected cognitive emotions, which in turn reduced job performance. Therefore, future studies can add mediating and moderating variables if performance is used as the object of study.

### 4.4. Expand Mediating and Moderating Variables

The literature from 2020–2022 further extends the research on mediating variables. In terms of employee perspective, Moqbel et al. [100] found that this behavior influenced intention to leave by enhancing two mediating variables, namely job satisfaction and also decreasing job tension. Saleem et al. [112] found that the excessive use caused relationship conflict, which reduced job performance. In terms of social interaction mechanisms, Yang et al. [95] found a mediating role of knowledge sharing. Sun et al. [113] found that features of social media enabled employees to knowledge acquisition and knowledge provision, which further promoted creativity. This is supported by the study of Jia et al. [110]. Wei et al. [91] found that employees’ use of social media can increase employee agility through meta-knowledge. In terms of organization perspective, Nivedhitha and Manzoor [99] concluded that this behavior can increase social connections to the workplace, thereby reducing cyber-slacking. Cai et al. [101] found that this behavior enhances employees’ interpersonal relationships and thus positively affects their performance. Employees’ use of social media increased leader–member communication, which stimulated employee creativity [93]. In addition, new mediating variables have been added to the literature in the last three years. One is organizational identity. This positively affects employee work engagement by increasing organizational identity [96,97,98]. The second is work stress. Liu et al. [83] found the mediating role of challenging and hindering pressures; i.e., challenging pressures positive mediate employees’ work enthusiasm, while hindering pressures are negatively mediating with employees’ work enthusiasm. The third variable is work engagement. Zhao et al. [108] found that this behavior increased work engagement and reduced work interruptions, thus achieving increased employee performance. Cheng et al. [94] found that this behavior stimulated work engagement, which in turn stimulated employees’ innovative performance. The fourth variable is psychological mechanisms; Ma et al. [106] found that this behavior affects employees’ emotional maintenance, which in turn affects their job performance; and Tang et al. [103] found that emotional exhaustion mediates the relationship between employees’ social media use and intention to leave their organization. Luqman et al. [114] concluded that this behavior causes disruption overload and psychological transition, which triggers exhaustion and loss of creativity. Nusrat et al. [104] found that this behavior induces certain psychological conditions that act as mediating variables to trigger cyber-slacking. There are other mediating variables. For example, Han and Xia [115] found the mediating role of employee voice for employee innovation. Cheng and Cho [84] found the mediating role of work break activities for employee recovery experience. Therefore, future studies can add new mediating variables based on these two aspects.

The 2020–2022 literature focuses less on moderating mechanisms and proposes new moderating variables. In addition to the positive moderating effect of task uncertainty on the relationship between social media use and job performance proposed by Lee and Lee [116] and in line with the task ambiguity mentioned above, some scholars have provided their own insights in terms of employees’ own factors, such as the effect of positive regulation of job expertise on employee agility [92], the effect of positive regulation of promotion on employee psychological transition, the effect of negative regulation of defense on employee psychological overload [114], and the effect of negative regulation of work stress on employee agility [92,117]. In terms of work situation factors, Zhang et al. [90] and Ma et al. [118] emphasized the moderating effect of organizational innovation culture on employee social capital, and Oksa et al. [119] explored the moderating effect of COVID-19 on employee work engagement. Wei et al. [91] argued that digital fluency in the organizational environment enhances employee agility. Tang et al. [103] found that work–life balance weakened employees’ intention to leave their jobs. Research on moderating mechanisms is relatively scarce but could be a focus for future research. Scholars can add moderating variables to fill the research gap.

### 4.5. Extend Theoretical Framework

The last three years of research from 2020–2022 are still based on the above theoretical framework but have made extensions. In terms of the mass communication perspective, Sun et al. [120] employed use and satisfaction theory by satisfying employees’ different needs. In terms of media richness, some scholars argue that the communication visibility and powerful information processing capabilities of social media have a significant impact on employees. Yang et al. [95] found the impact on employee productivity based on communication visibility theory. Pitafi and Ren [121] used communication visibility theory. Pitafi et al. [92], based on information processing theory, found that this behavior positively affects employee agility. In terms of personal motivation perspective, Zhao et al. [108], based on resource conservation theory, found that this behavior can increase work engagement and reduce work interruptions to influence employee performance. Luqman et al. [114] used resource conservation theory (COR) to investigate the role of interference overload and psychological transitions on employee fatigue and creativity. Pekkala and van Zoonen [76], based on social cognitive theory, found that perceived organizational commitment was an antecedent. Saleem et al. [112], based on social cognitive theory (SCT), found that the excessive use affects cognitive emotions and thus relational conflict, which in turn reduces job performance. Some scholars have added theories related to individual perspectives. Wei et al. [74] found from the self-determination theory three basic psychological needs of employees (i.e., need for competence, need for autonomy, and need for relatedness) influencing employees’ social media use. Zhou and Mou [81] found that this behavior positively influenced employees’ spirituality based on self-determination theory. Wu et al. [109], based on the transactional theory of stress, found that this behavior generates challenging work stress and hinders work stress, while challenging work stress helps employees to be more effective, and hindering work stress has a negative effect. Lee [80] studied employees’ motivation to use social media based on social ecological theory in terms of individual, interpersonal, and organizational factors. Tandon et al. [85] found negative effects of fear of missing out on employees’ psychology and behavior based on compensatory internet use theory and regulatory focus theory. In terms of interpersonal interaction perspectives, recent studies have expanded theories related to interpersonal interaction. Wang et al. [93], based on social exchange theory, found that employees’ use of social media stimulated leader–member exchange, which in turn stimulated employee creativity. Zhang et al. [90] discussed the moderating role of innovation culture using relational capital theory. Zhang et al. [79] used Guanxi theory to reveal the impact of social media communication function on employees. Nivedhitha and Manzoor [99] found that social ties in the workplace negatively affect employees’ network-slacking based on social bonding theory. Therefore, future studies can add theoretical frameworks and study from different perspectives, which may lead to more interesting conclusions.

## 5. Conclusions

In summary, with the widespread use of social media software in organizations, the study of how social media use in the workplace affects employee mental health and behavior has become an important research direction in the field of environment research and public health in recent years. This paper systematically summarizes and compares the research on employees’ social media use in recent years from the conceptual connotation, measurement methods, influencing factors, impact effects, mechanism of action, and theoretical perspectives and points out the contradictions and shortcomings of existing research. It also provides ideas and research directions for future research, namely the implementation research of multicultural background, the use of Python technology to crawl a large amount of data, and the text analysis method to analyze the psychological state of employees as well as the addition of new mediating variables, moderating variables, and consequence variables.

In terms of theoretical contributions, Chu [10] used meta-analysis to systematically analyze the articles on the relationship between social media use and employee outcomes from 2009 to 2018, while this paper focused on the literature after 2010. In particular, in the discussion section, related studies in the last three years from 2020 to 2022 were added, indicating the direction of future research for scholars. Sun et al. [11] mainly focused on the dark side of social media use, emphasizing the conflicts caused by employees’ use of social media, while this paper objectively evaluates the consequences of employees’ use of social media from both positive and negative aspects. van Zoonen et al. [12] classified related literature by individual behavior theory, social behavior theory, and communication theory, while this paper constructed a more comprehensive model based on the definition, types, research methods, post-effects mechanisms, mediating mechanisms, and moderating mechanisms, which clearly demonstrated the research context and frontier development in this field. In terms of practical contributions, the first is to provide a comprehensive model for scholars in this field, pointing out the key issues of current research and possible future research directions. Secondly, this paper provides theoretical support for whether relevant enterprises should allow employees to use social media.

Although the scoping review was comprehensive, analytical, and theoretical, the design of our scoping review prevented us from adequately testing the proposed theory because we only summarized and summarized papers published in the Social Science Citation Index. Therefore, future research should rely on more diverse literature sources and analysis methods to provide a more systematic and comprehensive analysis of employee social media use.

## Figures and Tables

**Figure 1 ijerph-19-16965-f001:**
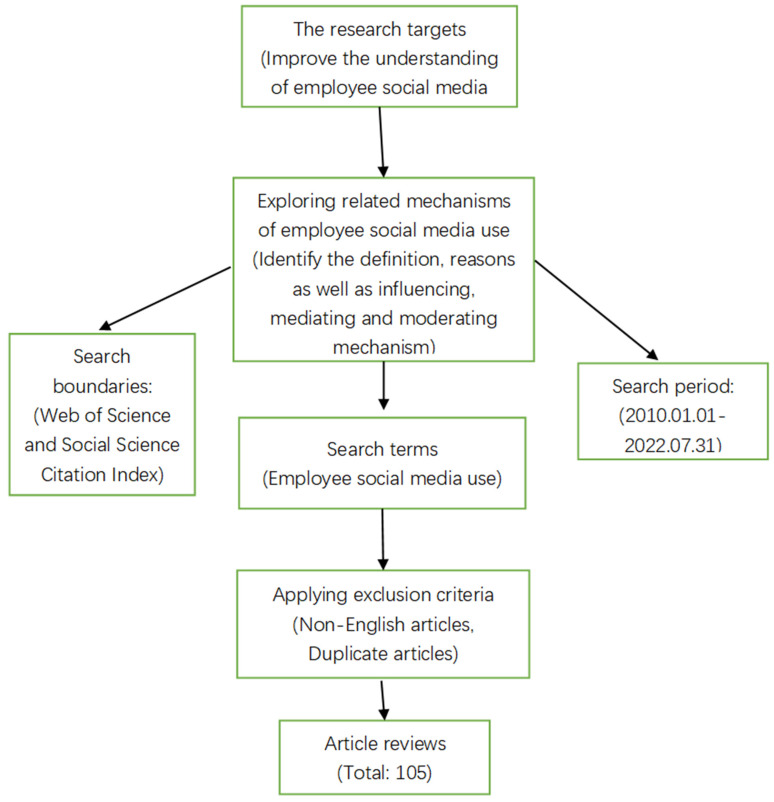
Research design for the scoping review.

## Data Availability

Not applicable.

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
