# Peer review of "The Impact of Social Media on Employee Mental Health and Behavior Based on the Context of Intelligence-Driven Digital Data"

_ijerph, 2022, doi:10.3390/ijerph192416965_

Round 1

Reviewer 1 Report

First, I would like to thank the editor and the authors for the opportunity to review the manuscript. Generally, the article is well written and addresses the understudied and current topic.

The aim of the study was to clarify the development and problems of employee social media use research through a systematic review. This study used a systematic literature review 60 method to sort out the definition, types, research methods, after-effects mechanisms, mediating mechanisms and moderating mechanisms of employee social media use. It is appropriately structured. However, there are some issues that would warrant further discussion.

I think that it would be reasonable to describe in more detail the difference between recreational entertainment, and work-based social media use. For example, it is not clear how authors categorize WhatsApp, zoom, and MS Teams work-based social media use. How they differentiate videoconference tools such as zoom, and MS Teams from recreational social media applications.

In the materials and methods section (page 2, lines 73-77) authors give some more justification for the given study that I think would be more suitable outlined in the background session.

It is not clear why authors decided to select only the Web of Science database (why they did not include other relevant databases) for study screening?

In the manuscript, there is a figure that depicts the evaluation procedure. However, there is missing important information (e.g., number of articles screened based on titles, full texts, reasons not to include for the final review).

There should be more information on inclusion and exclusion criteria used in the study. It is also not clear whether they screened the “grey literature” to get more studies in the study.

Results are discussed from multiple angles, and conclusions seem to answer the aims of the study. However, I think the major flaw of the manuscript is that the justification of the study is too “thin” in this form, and I suggest strengthening it.

Another weakness is related to the method approach used in the manuscript. To my understanding the style of this review is more like a scoping review than the systematic review. In this form it seems to include both elements. I suggest the authors follow either approach.

If authors decide to follow the systematic review process I suggest to follow rigorously PRISMA statement (Moher et al. 2009), or Preferred Reporting Items for Systematic Reviews and Meta-Analyses, they provide a checklist for review authors on how to report a systematic review.

This may be a piece constituting a new contribution to research after addressing the above issues. I hope these comments are useful.

Author Response

Response to Reviewer 1 Comments

Point 1: I think that it would be reasonable to describe in more detail the difference between recreational entertainment, and work-based social media use. For example, it is not clear how authors categorize WhatsApp, zoom, and MS Teams work-based social media use. How they differentiate videoconference tools such as zoom, and MS Teams from recreational social media applications.

Response 1: In terms of the difference between recreational entertainment and work-based social media use, I have classified the types of social media used by employees with two methods, namely work purpose and social purpose as well as public and private social media.

In terms of the purpose of use, there are work purpose and social purpose (Zhang et al., 2019).  Work purpose refers to this behavior mainly for work-related task activities; social purpose mainly refers to employees' use for entertainment, leisure, and other non-work-related activities.  Huang and Liu (2017) argue that social media use for work purposes can better improve employees' work efficiency, well-being, and job satisfaction. Social media use by employees for social purposes when working is often considered to be a massive waste of time and can reduce the productivity of the company (Yuan Sun et al., 2021). However, there are some studies that suggest that employees' socially-oriented social media use does not necessarily reduce organizational productivity. Song et al. (2019) showed that work-oriented and socially-oriented social media use are a complementary resource that can create synergistic effects and improve team and employee performance.

Social media can be classified into public and private social media. Public social media refers to the type of social media provided or identified by the organization in the workplace. Private social media refers to private social media used by employees when working. van Zoonen et al. (2017) suggest that public social media can increase the effectiveness of organizational communication and employee engagement at work and reduce fatigue. However, research has also shown that the use of public social media disrupts employees' attention, interrupts workflow, and increases work-life conflict (Gibbs et al., 2013). About private social media in the workplace, some scholars have argued that it can result in wasted employee time and reduced productivity (Leftheriotis & Giannakos, 2014). Yuan Sun et al. (2021), on the other hand, argue that there is no strong evidence supports that employees' use of private social media hinders work productivity; rather, many employees are able to maintain the energy and vitality needed to work effectively by using private social media to chat with family and friends, which can help them release the stress they face at work.

Point 2: In the materials and methods section (page 2, lines 73-77) authors give some more justification for the given study that I think would be more suitable outlined in the background session.

Response 2: About research justification, I add the detailed illustration in the introduction part.

There are many literature reviews about the impact of social media on employee mental health and behavior. Chu (2020) systematically analyzed the articles on the relationship between the use of social media and employee performance from 2009 to 2018 with the method of meta-analysis, but there are no latest studies from 2020 to 2022. Y. Sun et al. (2021) mainly focused on the dark side of social media use and emphasized the conflicts brought by employees' use of social media, but there are no reviews evaluating objectively the consequences of employees' use of social media from both positive and negative aspects. van Zoonen et al. (2022) classified relevant literature according to individual behavior theory, social behavior theory and communication theory, but no paper built a comprehensive model to clearly show the research background and frontier development in this field.

Point 3: It is not clear why authors decided to select only the Web of Science database (why they did not include other relevant databases) for study screening?

Response 3: The SSCI library of Web of Science is more rigorous in the peer review process, and the research papers covered are of high quality and high standards.

Point 4: In the manuscript, there is a figure that depicts the evaluation procedure. However, there is missing important information (e.g., number of articles screened based on titles, full texts, reasons not to include for the final review).

Response 4: The figure means that I searched academic papers about ‘Employee social media use’, published between 2010.01.01 and 2022.07.31, from Social Science Citation Index in Web of Science, excluding Non-English articles and Duplicate articles. The number of papers is 105.

Point 5: There should be more information on inclusion and exclusion criteria used in the study. It is also not clear whether they screened the “grey literature” to get more studies in the study.

Response 5: I searched academic papers about ‘Employee social media use’, published between 2010.01.01 and 2022.07.31, from Social Science Citation Index in Web of Science, excluding Non-English articles and Duplicate articles. The number of papers is 105. Not include grey literature.

Point 6: Results are discussed from multiple angles, and conclusions seem to answer the aims of the study. However, I think the major flaw of the manuscript is that the justification of the study is too “thin” in this form, and I suggest strengthening it.

Response 6: About research justification, I add the detailed illustration in the introduction part.

There are many literature reviews about the impact of social media on employee mental health and behavior. Chu (2020) systematically analyzed the articles on the relationship between the use of social media and employee performance from 2009 to 2018 with the method of meta-analysis, but there are no latest studies from 2020 to 2022. Y. Sun et al. (2021) mainly focused on the dark side of social media use and emphasized the conflicts brought by employees' use of social media, but there are no reviews evaluating objectively the consequences of employees' use of social media from both positive and negative aspects. van Zoonen et al. (2022) classified relevant literature according to individual behavior theory, social behavior theory and communication theory, but no paper built a comprehensive model to clearly show the research background and frontier development in this field.

Point 7: Another weakness is related to the method approach used in the manuscript. To my understanding the style of this review is more like a scoping review than the systematic review. In this form it seems to include both elements. I suggest the authors follow either approach.

Response 7: I choose systematic review method.

Point 8: If authors decide to follow the systematic review process I suggest to follow rigorously PRISMA statement (Moher et al. 2009), or Preferred Reporting Items for Systematic Reviews and Meta-Analyses, they provide a checklist for review authors on how to report a systematic review.

Response 8: The checklist is attached as word manuscript-supplementary.

Reviewer 2 Report

The manuscript has clear arguments, reasonable structure, and systematically expounds the ins and outs of research in this field, but there are still some deficiencies:

1. This manuscript uses a systematic literature review for this topic. Authors should justify their motivations using a systematic literature review.

2. The discussion section points out the latest research results for 2020-2022. It allows scholars in the field to gain insight into research frontiers, but what are the author's insights? It should be a critical contribution to this review paper.

3. This article systematically reviews the relevant literature on employee social media use from 2010-2022. Why choose 2010 at this point? Please provide specific reasons.

4. This article searches the SSCI repository from the Web of Science. There is a need to justify why chooses this repository.

5. This article has constructed a research framework, but it does not state the research contribution of this article. 

6. Suggestions for improvement are put forward in the last part of the paper, but limitations and future research venues also should be presented.

7. The authors need to proofread the logical flow of this manuscript.

Author Response

Response to Reviewer 2 Comments

Point 1: This manuscript uses a systematic literature review for this topic. Authors should justify their motivations using a systematic literature review.

Response 1: About research justification, I add the detailed illustration in the introduction part.There are many literature reviews about the impact of social media on employee mental health and behavior. Chu (2020) systematically analyzed the articles on the relationship between the use of social media and employee performance from 2009 to 2018 with the method of meta-analysis, but there are no latest studies from 2020 to 2022. Y. Sun et al. (2021) mainly focused on the dark side of social media use and emphasized the conflicts brought by employees' use of social media, but there are no reviews evaluating objectively the consequences of employees' use of social media from both positive and negative aspects. van Zoonen et al. (2022) classified relevant literature according to individual behavior theory, social behavior theory and communication theory, but no paper built a comprehensive model to clearly show the research background and frontier development in this field.

Point 2: The discussion section points out the latest research results for 2020-2022. It allows scholars in the field to gain insight into research frontiers, but what are the author's insights? It should be a critical contribution to this review paper.

Response 2: In terms of theoretical contributions, Chu (2020) used meta-analysis to systematically analyze the articles on the relationship between social media use and employee outcomes from 2009 to 2018, while this paper focused on the literatures after 2010. In particular, in the discussion section, related studies in the last three years from 2020 to 2022 were added, indicating the direction of future research for scholars. Y. Sun et al. (2021) mainly focuses on the dark side of social media use, emphasizing the conflicts caused by employees' use of social media, while this paper objectively evaluates the consequences of employees' use of social media from both positive and negative aspects. van Zoonen et al. (2022) classified related literature by individual behavior theory, social behavior theory and communication theory, while this paper constructed a more comprehensive model based on the definition, types, research methods, post-effects mechanisms, mediating mechanisms and moderating mechanisms, which clearly demonstrated the research context and frontier development in this field. In terms of practical contributions, the first is to provide a comprehensive model for scholars in this field, pointing out the key issues of current research and possible future research directions. Secondly, it provides theoretical support for whether relevant enterprises allow employees to use social media.

Point 3: This article systematically reviews the relevant literature on employee social media use from 2010-2022. Why choose 2010 at this point? Please provide specific reasons.

Response 3: In the part of definition, Aichner et al. (2020) mentioned that the definition of social media changed dramatically in 2010. Previously, social media was a communication platform used by a group of people with common interests to share and deliver information, whereas afterwards a large empirical literature focused on the idea that social media can create and share user-generated content (Kaplan & Haenlein, 2014), which implies a change in the focus of social media research. The results echo the findings regarding the progression of social media research that the definition of social media has changed over time, from a social platform in the past to a tool for information aggregation (Kapoor et al., 2018). Because the definition of social media changed in 2010, articles after 2010 are selected for review.

Point 4: This article searches the SSCI repository from the Web of Science. There is a need to justify why chooses this repository.

Response 4: The SSCI library of Web of Science is more rigorous in the peer review process, and the research papers covered are of high quality and high standards.

Point 5: This article has constructed a research framework, but it does not state the research contribution of this article.

Response 5: In terms of theoretical contributions, Chu (2020) used meta-analysis to systematically analyze the articles on the relationship between social media use and employee outcomes from 2009 to 2018, while this paper focused on the literatures after 2010. In particular, in the discussion section, related studies in the last three years from 2020 to 2022 were added, indicating the direction of future research for scholars. Y. Sun et al. (2021) mainly focuses on the dark side of social media use, emphasizing the conflicts caused by employees' use of social media, while this paper objectively evaluates the consequences of employees' use of social media from both positive and negative aspects. van Zoonen et al. (2022) classified related literature by individual behavior theory, social behavior theory and communication theory, while this paper constructed a more comprehensive model based on the definition, types, research methods, post-effects mechanisms, mediating mechanisms and moderating mechanisms, which clearly demonstrated the research context and frontier development in this field. In terms of practical contributions, the first is to provide a comprehensive model for scholars in this field, pointing out the key issues of current research and possible future research directions. Secondly, it provides theoretical support for whether relevant enterprises allow employees to use social media.

Point 6: Suggestions for improvement are put forward in the last part of the paper, but limitations and future research venues also should be presented.

Response 6: Although systematic literature review was comprehensive, analytical, and theoretical, the design of our systematic literature review prevented us from adequately testing the proposed theory because we only summarized and summarized papers published in the Social Science Citation Index. Therefore, future research should rely on more diverse literature sources and analysis methods to provide a more systematic and comprehensive analysis of employee social media use.

This paper provides ideas and research directions for future research, namely, the implementation research of multicultural background, the use of Python technology to crawl a large amount of data and the text analysis method to analyze the psychological state of employees, as well as the addition of new mediating variables, moderating variables and consequence variables.

Point 7: The authors need to proofread the logical flow of this manuscript.

Response 7: The logical flow have been proofread.

Round 2

Reviewer 1 Report

I would like to thank the authors for the careful revisions of the manuscript.

I am mainly happy with the corrections. I still suggest considering to change the focus of the study on scoping review instead of the systematic literature review.

Author Response

Response to Reviewer 1 Comments

Point 1: I would like to thank the authors for the careful revisions of the manuscript.

I am mainly happy with the corrections. I still suggest considering to change the focus of the study on scoping review instead of the systematic literature review.

Response 1: Thanks for your comments.Now, I have changed the research methodology from systematic literature review to scoping review.

Reviewer 2 Report

Thanks for addressing my concerns. I can tell that there is a big improvement in the current form, well done.

Author Response

Response to Reviewer 2 Comments

Point 1: Thanks for addressing my concerns. I can tell that there is a big improvement in the current form, well done.

Response 1: Thanks for your comments.Now, my article has improved a lot.
